# Biomimetic Remineralization Strategies for Dentin Bond Stability—Systematic Review and Network Meta-Analysis

**DOI:** 10.3390/ijms26083488

**Published:** 2025-04-08

**Authors:** Rosário Costa, Joana Reis-Pardal, Sofia Arantes-Oliveira, João Cardoso Ferreira, Luis Filipe Azevedo, Paulo Melo

**Affiliations:** 1Faculty of Dental Medicine, Department of Operative Dentistry, University of Porto, Rua Dr. Manuel Pereira da Silva, 4200-393 Porto, Portugal; jcferreira@fmd.up.pt (J.C.F.); pmelo@fmd.up.pt (P.M.); 2CINTESIS@RISE—Center for Health Technology and Services Research (CINTESIS), Health Research Network Associated Laboratory (RISE), University of Porto, 4200-450 Porto, Portugal; jrpardal@med.up.pt (J.R.-P.); lazevedo@med.up.pt (L.F.A.); 3Department of Community Medicine, Information and Health Decision Sciences (MEDCIS), Faculty of Medicine, University of Porto, Alameda Prof. Hernâni Monteiro, 4200-319 Porto, Portugal; 4Department of Dental Biomaterials, Faculty of Dental Medicine, University of Lisbon, Cidade Universitária, Rua Prof.ª Teresa Ambrósio, 1600-277 Lisbon, Portugal; sofiaaol@campus.ul.pt; 5EpiUnit, ITR, Institute of Public Health, University of Porto, Rua das Taipas, n° 135, 4050-600 Porto, Portugal

**Keywords:** adhesives, biomimetic material, dentin-bonding agents, tooth demineralization, tooth remineralization

## Abstract

This systematic review and network meta-analysis aimed to evaluate the bond strength of artificial caries-affected dentin (ACAD) of permanent human teeth with and without biomimetic remineralization (BR), assessed based on in vitro studies. Following PRISMA guidelines, we conducted a systematic search until June 2023, identifying 82 eligible articles for full-text analysis. We assessed the study characteristics, methodological quality, and summary results. Bond strength was examined immediately and after artificial aging using three bond strength tests. We performed meta-regressions (using OpenBUGS software) to explore the relationship between the independent variable’s adhesive application technique (Etch-and-Rinse or Self-Etch) and ACAD protocol (chemical or biological) and the dependent variable of bond strength. Additionally, we conducted random-effect NMAs (using CINEMA software) to compare the effect of multiple interventions per application technique and ACAD protocol simultaneously. Among the included studies that compared various BR strategies, most studies (19 out of 22) presented a medium risk of bias. In some comparisons, the meta-regression results revealed a significant association between bond strength at 24 h and both the adhesive application technique and the ACAD protocol. Our findings indicate the potential of BR to enhance bond strength in human ACAD in in vitro settings.

## 1. Introduction

Dentin-bonding procedures pose persistent challenges in Operative Dentistry despite the currently significant successes achieved in enamel bonding [1]. A well-documented issue in the literature is the gradual deterioration of the adhesive systems’ bond strength to dentin over time, primarily due to hybrid layer degradation [2]. This compromise in dentin bonding significantly limits the lifespan of adhesive restorations [3].

The ideal dentin-bonding process involves exposing the collagen network and facilitating the penetration of chelating agents or acidic functional monomers to form the crucial hybrid layer [4]. However, a portion of the exposed collagen matrix remains unfilled with resin monomers, rendering it susceptible to hydrolytic degradation over time, thus jeopardizing the longevity of dentin bonding due to nanoleakage. The incomplete water removal within hydrophilic resin monomers also creates a weak point in resin-dentin bonds [5,6]. These phenomena have led to the exploration of an innovative approach to improve dentin adhesion: the biomimetic remineralization (BR) of collagen fibrils exposed during biomineralization [7,8].

There are two primary BR strategies: incorporating mineral-promoting agents into adhesives or restorative materials and applying pre-treating solutions before adhesive systems [9,10]. For the first strategy, researchers have developed experimental adhesive systems or restorative materials containing bioactive components like calcium phosphate or other inorganic materials that supply mineral ions to remineralize the resin–dentin interface [11,12]. The second strategy involves solutions containing non-collagenous proteins or template analogs to stimulate intra/extra-fibrillar mineralization [13,14]. These remineralizing agents facilitate the formation of nanometric apatite crystals, which replace excess water, mimicking physiological remineralization [14], thus enhancing the structural integrity of dentin and extending the longevity of the dentin–composite resin bonding interface [7,15,16]. Some studies have also suggested that these agents can inhibit the degradation of exposed collagen by attracting calcium to it [17].

Therefore, it is essential to analyze the challenges posed by dentin-bonding procedures and the potential advantages of BR procedures. This systematic review uses a comprehensive network meta-analysis (NMA) to assess and compare the bond strength of human artificial caries-affected dentin (ACAD) with and without BR evaluated in in vitro studies.

## 2. Materials and Methods

### 2.1. Search Strategy

This systematic review was registered in PROSPERO and performed according to the PRISMA statement [18]. On June 2023, PubMed, ISI Web of Science, and SCOPUS were searched to identify potentially relevant studies. In addition to electronic databases, reference lists of included studies and relevant systematic reviews were also searched. The complete search strategies are available in Appendix A.

### 2.2. Outcomes

The primary outcome of this systematic review was determining the mean difference between the bond strength of ACAD with and without BR using different adhesive application techniques, including Etch-and-Rinse (ER) or Self-Etch (SE), and ACAD protocols, including chemical or biological.

### 2.3. Eligibility Criteria

The following inclusion criteria were established: experimental or quasi-experimental in vitro studies investigating the influence of any BR procedure on the ACAD–adhesive interface’s bond strength; having a control group (dentin without BR) for comparison; ACAD protocols in which agents were applied immediately prior to bonding; outcomes measured based on shear, micro-shear, or micro-tensile bond strength (SBS, µSBS, µTBS) tests. The exclusion criteria included studies with doped materials or modified adhesive systems.

The terms “caries-affected dentin”, “demineralized dentin”, and “artificial eroded dentin” were considered as references to ACAD. ACAD consists of human dentin tissue artificially demineralized to mimic the characteristics of dentin affected by carious changes. It is created by exposing dentin tissue to acidic or demineralizing solutions to remove mineral content, leading to softening and structural alterations like those observed in natural caries-affected dentin [19,20,21]. This demineralization process is performed in a laboratory setting to replicate the conditions and properties of carious dentin.

The BR procedures considered included any technique aimed at restoring and strengthening damaged or demineralized dentin in a way that mimicked the tooth’s natural remineralization process [3,22].

### 2.4. Data Extraction and Collection

Firstly, two authors (RC and JP) independently reviewed titles and abstracts to select articles for further assessment per their consensus. Disagreements were resolved by discussion until a consensus was reached. Full texts of the selected articles were retrieved, and the same two authors further evaluated and independently extracted data from them. The reference lists of the included full texts were also screened and cross-referred.

In the case of missing/unclear items (e.g., missing bond strength measurements, missing standard deviation values, uncertain number of samples used) or inconsistent data within or between sources (e.g., differences in data between text and figures, bond strength measurements only in figures), the authors of the respective studies were contacted via e-mail. Two follow-up e-mails were sent with a one-week interval.

The search results from the online databases were imported to Endnote20 (Clarivate, Philadelphia, USA), where duplicates were removed. The Rayyan app [23] was used to keep records and assist in abstract screening, full-text review, and data extraction. Data for the systematic review and NMA were extracted using a custom-made Excel worksheet.

The following items were extracted from each source: authors; year of publication; study randomization; risk of bias; means and standard deviations; number of samples; ACAD protocol (chemical or biological); BR procedure; adhesive type used (ER, SE, or universal) and adhesive application technique; method of bond strength assessment; outcome measurement time point (24 h or after artificial aging method).

The authors classified and grouped the treatments by active substance into nine groups: fluorine, calcium phosphate, peptide, silica, hydroxyapatite, flavonoids, calcium, and 2-hydroxyethyl methacrylate/ethylene glycol dimethacrylate (HEMA/EDGMA).

### 2.5. Risk of Bias Assessment

Two authors (RC and JP) independently assessed the risk of bias in the included in vitro studies according to the QUIN tool [24]. Disagreements were resolved by discussion until a consensus was reached. Each study was graded accordingly as having high, medium, or low risk based on the final score of the tool: low risk of bias if >70%, medium risk of bias if 50–70%, and high risk of bias if <50%.

### 2.6. Data Synthesis and Statistical Analysis

#### 2.6.1. Qualitative Synthesis

Qualitative evidence synthesis was performed via descriptive analysis of the studies’ characteristics, methodologic quality, and summary results using a narrative description and summary tables, providing a clear overview of the individual study characteristics, main findings, and methodological assessments.

#### 2.6.2. Quantitative Synthesis

Quantitative syntheses were performed via random-effects NMA of the mean difference between the intervention and control groups. NMAs were conducted using the CINEMA software (https://cinema.ispm.unibe.ch/), based on the R software (https://www.r-project.org/) packages meta an netmeta [25,26], using the adhesive application technique and ACAD protocol and including all possible pair-wise comparisons based on direct and indirect evidence. In accordance with the Cochrane guidelines [27], when trials had more than two arms, we combined interventions into a single group if they belonged to the same intervention category. When more than one independent treatment–comparator pair existed in each study, we treated them as if they pertained to independent studies. Following the Cochrane guidelines [27], standard deviations were imputed from other included studies in cases where they were not available in the manuscript and could not be obtained upon contact with the authors.

The rating of confidence in the results was assessed following the CINEMA approach by evaluating the following domains: within-study bias, reporting bias, indirectness, imprecision, heterogeneity, and incoherence. The minimal clinically important difference was established by consensus of the authors as 7 megapascals (MPa).

In addition, since it has been reported that the adhesive application technique (ER vs. SE) [8,25,26,27] and the ACAD protocol (chemical vs. biological) [7,28] might influence the BR treatment’s effect, we explored the effects of these two covariates in NMA effects estimates based on random-effects Bayesian meta-regressions using the OpenBUGS software version 3.2.3 (Code in Appendix A). Within a random-effects Bayesian framework, the OpenBUGS software [28] was also used to estimate each intervention’s posterior median ranks and probability to be the best.

Finally, to assess the robustness of the results obtained from NMAs, as assumptions change, we conducted the following two sensitivity analyses:Random selection of one treatment intervention: Instead of combining interventions belonging to the same intervention category, as in the main analysis, we randomly selected only one.Removal of SBS test results: Instead of including all bond strength tests, as in the main analysis, we included only results from µSBS and µTBS tests.

## 3. Results and Discussion

### 3.1. Search Results

In the electronic search, 1874 records were identified after eliminating duplicates. Only 82 were selected for full-text screening. The reasons for the exclusion of screened full texts are shown in Appendix A Table A1. After critical appraisal, 23 remaining articles were included in our systematic review and 22 in the NMA. A PRISMA flow diagram of the complete process is illustrated in Figure 1.

### 3.2. Characteristics of Included Studies

Table 1 displays the characteristics of the included studies, interventions, and outcomes. Of the 23 studies in the systematic review, 16 were experimental [8,15,29,30,31,32,33,34,35,36,37,38,39,40,41,42] and 7 were quasi-experimental [7,14,43,44,45,46,47]. One study was excluded from the NMA because it lacked reporting data, which could not be obtained upon direct contact with the authors (Appendix A Table A2, Table A3 and Table A4).

All 22 studies in the NMA performed immediate (24 h) bond strength measurements. Of these studies, 13 investigated the ER technique associated with the chemical ACAD protocol [7,8,29,30,31,33,34,35,38,39,40,46], 5 investigated the ER with the biological ACAD [7,15,32,37,44], 13 investigated the SE with the chemical ACAD [8,14,31,33,34,35,36,41,43,45,47,48,49], and only 1 investigated the SE with the biological ACAD [37]; the latter was insufficient to perform an NMA. In turn, 11 studies measured bond strength after artificial aging of the specimens: 4 used thermocycling [14,32,38,50], and 7 stored them in a fluid solution for months [15,29,39,40,43,44,47].

Overall, both immediate and aged bond strength in the ACAD benefited from BR. The artificial aging method globally diminished bond strength values, and thermocycling caused the lowest bond strength.

**Table 1 ijms-26-03488-t001:** Characteristics of the included studies, interventions, and outcomes.

	Study/Year	RoB (Score)	Study Type	ACAD	BRP	Groups	N (Teeth)	Mean (SD)	AT	OM Test
24 h measurement										
ER + C	Altinci et al., 2018 [40]	M (50)	Exp.	32% phosphoric acid	Control	Control	9	35.27 (4.63) ^a^	ER	µTBS
					F	NaF + 6 mM F		34.7 (4.63) ^a^		
						NaF + 24 mM F		54.66 (4.63) ^a^		
						NaF + 179 mM F		47.11 (4.63) ^a^		
						KF + 6 mM F		51.8 (4.63) ^a^		
						KF + 24 mM F		48.56 (4.63) ^a^		
						KF + 179 mM F		47.58 (4.63) ^a^		
						CaF2 + 6 mM F		36.34 (4.63) ^a^		
						CaF2 + 24 mM F		39.49 (4.63) ^a^		
						CaF2 + 179 mM F		48.47 (4.63) ^a^		
						Excite F		48.84 (4.63) ^a^		
	Barbosa-Martins et al. (A) 2018 [8]	M (54)	Exp.	6% CMC	Control	Control	6	26.38 (8.64)	ER	µTBS
					F	NaF		33.43 (10.41)		
					CaP	CPP-ACP		45.25 (8.82)		
					Pept.	P11-4		46.42 (12.03)		
	Barbosa-Martins et al. (B) 2018 [7]	M (54)	Quasi-Exp.	6% CMC	Control	Control	6	21.96 (5.92)	ER	µTBS
					F	NaF		33.43 (10.42)		
					CaP	CPP-ACP		45.25 (8.83)		
					Pept.	P11-4		46.42 (12.03)		
	Bauer et al., 2018 [29]	M (50)	Exp.	35% phosphoric acid	Control	Control	13	17 (4.1)	ER	SBS
					CaP	5% NbG		17.9 (5)		
						10%NbG		15.8 (6.4)		
						20%NbG		16.6 (4.4)		
						40%NbG		15.8 (4.1)		
	Cardenas et al., 2021 [30]	M (63)	Exp.	pH cycling	Control	Control	5	33.74 (3.6)	Univ.	µTBS
					F	SDF 12%		38.03 (3.5)		
						SDF 38%		39.68 (2.7)		
						SDF 38% without KI		39.38 (2.5)		
					Control	Control		34.9 (3.3)		
					F	SDF 12%		42.45 (2.9)		
						SDF 38%		40.47 (4.2)		
						SDF 38% without KI		41.3 (2.5)		
	Chen et al., 2020 ^c^	M (54)	Quasi-Exp.	pH cycling	Control	Control	4	13.8 (3.35) ^a^	Univ.	µTBS
					CaP	Ca/P-PILP		23.8 (3.35) ^a^		
					Pept.	PAA-PASP		14 (3.35) ^a^		
					CaP	Ca/P		11.9 (3.35) ^a^		
	Cifuentes-Jimenez et al., 2021 [31]	M (50)	Exp.	pH cycling	Control	Control	5	31.4 (4.63) ^a^	ER	µTBS
					F	Cariestop		15.1 (4.63) ^a^		
						RivaStar1		10.1 (4.63) ^a^		
						RivaStar2		7.5 (4.63) ^a^		
						Saforide		23.2 (4.63) ^a^		
	Gungormus et al., 2021 [33]	M (50)	Exp.	37% phosphoric acid	Control	Control	10	15.38 (1.3)	ER	SBS
					CaP	NPR 60 min		15.85 (1.44)		
					Pept.	PR 10 min		20.81 (1.74)		
						PR 30 min		20 (1.68)		
						PR 60 min		16.21 (1.1)		
	Krithi et al., 2020 [34]	M (54)	Exp.	0.5% citric acid	Control	Control	15	11.83 (0.43)	ER	µSBS
					F	NaF		11.56 (0.15)		
					CaP	CPP-ACP		12.12 (0.57)		
						Novamin		11.66 (0.28)		
					Ca	Non-Fidated		11.94 (0.27)		
	Meng et al., 2021 [35]	M (50)	Exp.	1% citric acid	Control	Control	8	46.8 ^b^ (4.63) ^a^	Univ.	µTBS
					Hap	Biorepair		50.72 ^b^ (4.63) ^a^		
						Dontodent Sensitive		50.71 ^b^ (4.63) ^a^		
						nHAp		51.24 ^b^ (4.63) ^a^		
					Control	Control		50.41 ^b^ (4.63) ^a^		
					Hap	Biorepair		53.38 ^b^ (4.63) ^a^		
						Dontodent Sensitive		54.5 ^b^ (4.63) ^a^		
						nHAp		55.63 ^b^ (4.63) ^a^		
					Control	Control		46.85 ^b^ (4.63) ^a^		
					Hap	Biorepair		50.77 ^b^ (4.63) ^a^		
						Dontodent Sensitive		53.82 ^b^ (4.63) ^a^		
						nHAp		55 ^b^ (4.63) ^a^		
	Pulidindi et al., 2021 [38]	M (63)	Exp.	37% phosphoric acid	Control	Control	15	48.84 (4.63) ^a^	ER	µTBS
					Pept.	P11-4		38.66 (4.63) ^a^		
					CaP	CPP-ACP		34.07 (4.63) ^a^		
					Control	Control		22.63 (4.63) ^a^		
					Pept.	P11-4		25.37 (4.63) ^a^		
					CaP	CPP-ACP		23.62 (4.63) ^a^		
	Van Duker et al., 2019 [46]	H (46)	Quasi-Exp.	7 days in ADS	Control	Control	10	23.5 (10.7)	Univ.	µTBS
					F	SDF 38%		19.8 (8.4)		
						SDF 38% without KI		7.9 (6.6)		
	Yang et al., 2018 [39]	M (50)	Exp.	1% citric acid	Control	Control	10	46.5 ^b^ (4.63) ^a^	ER	µTBS
					CaP	CPP-ACP		42.6 ^b^ (4.63) ^a^		
						Novamin		43.3 ^b^ (4.63) ^a^		
					Control	Control		22.3 ^b^ (4.63) ^a^		
					CaP	CPP-ACP		41.2 ^b^ (4.63) ^a^		
						Novamin		31.4 ^b^ (4.63) ^a^		
ER + B	Barbosa-Martins et al. (B) 2018	M (54)	Quasi-Exp.	BHI+ S.Mutans	Control	Control	6	22.89 (2.68)	ER	µTBS
					F	NaF		26.94 (6.7)		
					CaP	CPP-ACP		47.95 (6.69)		
					Pept.	P11-4		42.07 (7.83)		
	Dávila-Sánchez et al., 2020 [32]	M (54)	Exp.	Cariogenic + S. Mutans	Control	Control	7	14.42 (4.43)	Univ.	µTBS
					Fls.	QUE		24.58 (4.9)		
						HES		18.41 (5.3)		
						RUT		26 (5.51)		
						NAR		24.64 (3.7)		
						PRO		20.66 (3.92)		
	de Sousa et al., 2019 [44]	M (50)	Quasi-Exp.	Cariogenic + S. Mutans	Control	Control	8	21.07 (3.24)	ER	µTBS
					Pept.	P11-4		42.07 (7.83)		
	Moreira et al., 2021 [15]	M (54)	Exp.	Cariogenic + S. Mutans	Control	Control	8	25.4 (2.45)	ER	µTBS
					F	NaF		25.47 (4.8)		
					CaP	CPP-ACP		41.79 (5.85)		
					Pept.	P11-4		40.12 (3.62)		
	Siqueira et al., 2020 [37]	M (63)	Exp.	Cariogenic + S. Mutans	Control	Control	5	16.81 (3.5)	Univ.	µTBS
					F	SDF 12%		21.11 (4.1)		
						SDF 38%		24.36 (3.4)		
					Control	Control		19.89 (2.4)		
					F	SDF 12%		24.47 (3.4)		
						SDF 38%		26.32 (2)		
SE + C	Atomura et al., 2018 [43]	H (46)	Quasi-Exp.	7 days in ADS	Control	Control	unknown	48.3 (13)	SE	µTBS
					F	NaF		47.7 (8.6)		
						FCP complex		43.9 (14.3)		
	Barbosa-Martins et al. (A) 2018 [8]	M (54)	Exp.	48 h 6% CMC	Control	Control	6	25.38 (8.58)	SE	µTBS
					F	NaF		35.59 (9.18)		
					CaP	CPP-ACP		48.11 (11.71)		
					Pept.	P11-4		25.7 (8.95)		
	Cardenas et al., 2021 [30]	M (63)	Exp.	pH cycling	Control	Control	5	33.74 (3.6)	Univ.	µTBS
					F	SDF 12%		39.53 (4.2)		
						SDF 38%		41.31 (2)		
						SDF 38% without KI		40.55 (2.9)		
					Control	Control		36.56 (4.1)		
					F	SDF 12%		39.98 (1.7)		
						SDF 38%		41.08 (3)		
						SDF 38% without KI		41.57 (2.4)		
	Chen et al., 2020 [14]	M (54)	Quasi-Exp.	pH cycling	Control	Control	4	13.8 (3.35) ^a^	Univ.	µTBS
					CaP	Ca/P-PILP		23.8 (3.35) ^a^		
					Pept.	PAA-PASP		14 (3.35) ^a^		
					CaP	Ca/P		11.9 (3.35) ^a^		
					Control	Control		9.2 (3.35) ^a^		
					CaP	Ca/P-PILP		15.1 (3.35) ^a^		
					Pept.	PAA-PASP		9.3 (3.35) ^a^		
					CaP	Ca/P		9.8 (3.35) ^a^		
	Cifuentes-Jimenez et al., 2021 [31]	M (50)	Exp.	pH cycling	Control	Control	5	31.4 (3.35) ^a^	SE	µTBS
					F	Cariestop		9.6 (3.35) ^a^		
						Saforide		8.03 (3.35) ^a^		
	Gungormus et al., 2021 [33]	M (50)	Exp.	37% phosphoric acid	Control	Control	10	15.38 (1.3)	SE	SBS
					CaP	NPR 60 min		15.49 (1.17)		
					Pept.	PR 10 min		18.93 (0.99)		
						PR 30 min		19.62 (0.9)		
						PR 60 min		21.73 (1.57)		
	Krithi et al., 2020 [34]	M (54)	Exp.	0.5% citric acid	Control	Control	15	11.83 (0.43)	SE	µSBS
					F	NaF		12.4 (0.18)		
					CaP	CPP-ACP		11.97 (0.39)		
						Novamin		11.97 (0.17)		
					Ca	Non-Fidated		10.62 (0.11)		
	Meng et al., 2021 [35]	M (50)	Exp.	1% citric acid	Control	Control	8	46.8 ^b^ (3.35) ^a^	Univ.	µTBS
					Hap	Biorepair		47.62 ^b^ (3.35) ^a^		
						Dontodent Sensitive		51.89 ^b^ (3.35) ^a^		
						nHAp		51.89 ^b^ (3.35) ^a^		
					Control	Control		56.3 ^b^ (3.35) ^a^		
					Hap	Biorepair		51.62 ^b^ (3.35) ^a^		
						Dontodent Sensitive		57.47 ^b^ (3.35) ^a^		
						nHAp		58.39 ^b^ (3.35) ^a^		
					Control	Control		56.8 ^b^ (3.35) ^a^		
					Hap	Biorepair		52.25 ^b^ (3.35) ^a^		
						Dontodent Sensitive		50.8 ^b^ (3.35) ^a^		
						nHAp		56.1 ^b^ (3.35) ^a^		
	Paik et al., 2022 [42]	M (50)	Exp.	35% phosphoric acid	Control	Control	4	21.66 (3.35) ^a^	Univ.	µTBS
					Fls.	ICT		24.4 (3.35) ^a^		
						FIS		26.81 (3.35) ^a^		
						SIB		25.65 (3.35) ^a^		
						CPIC		25.97 (3.35) ^a^		
						ICT + C		30.63 (3.35) ^a^		
						FIS + C		25.63 (3.35) ^a^		
						SIB + C		24.76 (3.35) ^a^		
	Pei et al., 2019 [36]	M (50)	Exp.	1% citric acid	Control	Control	4	43.61 (3.35) ^a^	SE	µTBS
					Hap	Biorepair		33.16 (3.35) ^a^		
						Dontodent Sensit.		35.41 (3.35) ^a^		
						nHAp		46.92 (3.35) ^a^		
					Control	Control		47.47 (3.35) ^a^		
					Hap	Biorepair		43.47 (3.35) ^a^		
						Dontodent Sensit.		42.3 (3.35) ^a^		
						nHAp		41.24 (3.35) ^a^		
	Priya et al., 2020 [45]	H (46)	Quasi-Exp.	37% phosphoric acid	Control	Control	13	6.677 (1.254)	Univ.	SBS
					F	VivaSens		3.332 (0.78)		
						MS Coat F		3.127 (0.478)		
					HEMA	GLUMA Desensit.		4.572 (0.718)		
						Systemp		9.697 (1.127)		
	Zang et al., 2018 [41]	M (50)	Exp.	37% phosphoric acid	Control	Control	6	19.73 ^b^ (2.108)	Univ.	SBS
					SiO_2_	Charged mesoporous		20.57 ^b^ (2.244)		
	Zumstein et al., 2018 [47]	M (50)	Quasi-Exp.	pH cycling	Control	Control	20	24.7 (8.1) ^c^	SE	µTBS
					F	SnCl_2_/AmF4		23.3 (8.2) ^c^		
					Control	Control		23.73 (8) ^c^	Univ.	
					F	SnCl_2_/AmF4		21.39 (6.8) ^c^		
SE + B	Siqueira et al., 2020 [37]	M (63)	Exp.	Cariogenic + S. Mutans	Control	Control	5	16.81 (3.5)	Univ.	µTBS
					F	SDF 12%		20.02 (4.6)		
						SDF 38%		25.21 (3)		
					Control	Control		19.61 (3.3)		
					F	SDF 12%		23.82 (4.4)		
						SDF 38%		27.16 (3.6)		
TMC measurement										
ER + C	Pulidindi et al., 2021 [38]	M (63)	Exp.	37% phosphoric acid	Control	Control	15	48.84 (4.63) ^a^	ER	µTBS
					Pept.	P11-4		25.37 (4.63) ^a^		
					CaP	CPP-ACP		23.62 (4.63) ^a^		
ER + B	Dávila-Sánchez et al., 2020 [32]	M (54)	Exp.	Cariogenic + S. Mutans	Control	Control	7	14.42 (4.43)	Univ.	µTBS
					Fls.	QUE		12.02 (5.21)		
						HES		15.73 (6.07)		
						RUT		21.08 (4.75)		
						NAR		22.12 (2.92)		
						PRO		17.2 (2.72)		
SE + C	Chen et al., 2020 [14]	M (54)	Quasi-Exp.	pH cycling	Control	Control	4	13.8 (3.35) ^a^	Univ.	µTBS
					CaP	Ca/P-PILP		15.1 (3.35) ^a^		
					Pept.	PAA-PASP		9.3 (3.35) ^a^		
					CaP	Ca/P		9.8 (3.35) ^a^		
	Paik et al., 2022 [42]	M (50)	Exp.	35% phosphoric acid	Control	Control	4	21.66 (3.35) ^a^	Univ.	µTBS
					Fls.	ICT		20.53 (3.35) ^a^		
						FIS		19.4 (3.35) ^a^		
						SIB		22.04 (3.35) ^a^		
						CPIC		23.43 (3.35) ^a^		
						ICT + C		26.74 (3.35) ^a^		
						FIS + C		23.42 (3.35) ^a^		
						SIB + C		25.17 (3.35) ^a^		
Storage in a fluid solution for 3-month measurement										
ER + C	Bauer et al., 2018 [29]	M (50)	Exp.	35% phosphoric acid	Control	Control	13	17 (4.1)	ER	SBS
					CaP	5% NbG		11.8 (3.7)		
						10%NbG		13.9 (3.2)		
						20%NbG		13.2 (2.7)		
						40%NbG		14.7 (2.9)		
SE + C	Atomura et al., 2018 [43]	H (46)	Quasi-Exp.	7 days in ADS	Control	Control	unknown	48.3 (13)	SE	µTBS
					F	NaF		42.6 (12.1)		
						FCP complex		47.4 (9.2)		
Storage in a fluid solution for 6-month measurement										
ER + C	Altinci et al., 2018 [40]	M (50)	Exp.	32% phosphoric acid	Control	Control	9	35.27 (4.63) ^a^	ER	µTBS
					F	NaF + 6 mM F		50.31 (4.63) ^a^		
						NaF + 24 mM F		49.28 (4.63) ^a^		
						NaF+179 mM F		47.73 (4.63) ^a^		
						KF + 6 mM F		41.95 (4.63) ^a^		
						KF + 24 mM F		51.53 (4.63) ^a^		
						KF + 179 mM F		54.29 (4.63) ^a^		
						CaF2 + 6 mM F		52.25 (4.63) ^a^		
						CaF2+24 mM F		41.1 (4.63) ^a^		
						CaF2+179 mM F		40.85 (4.63) ^a^		
						Excite F		46.22 (4.63) ^a^		
	de Sousa et al., 2019 [44]	M (50)	Quasi-Exp.	Cariogenic + S. Mutans	Control	Control	8	21.07 (3.24)	ER	µTBS
					Pept.	P11-4		31.98 (3.44)		
	Moreira et al., 2021 [15]	M (54)	Exp.	Cariogenic + S. Mutans	Control	Control	8	25.4 (2.45)	ER	µTBS
					F	NaF		18.36 (5.5)		
					CaP	CPP-ACP		36.55 (4.27)		
Storage in a fluid solution for 12-month measurement										
ER + C	Altinci et al., 2018 [40]	M (50)	Exp.	32% phosphoric acid	Control	Control	9	35.27 (4.63) ^a^	ER	µTBS
					F	NaF + 6 mM F		51.63 (4.63) ^a^		
						NaF + 24 mM F		45.56 (4.63) ^a^		
						NaF + 179 mM F		39.31 (4.63) ^a^		
						KF + 6 mM F		40.01 (4.63) ^a^		
						KF + 24 mM F		51.85 (4.63) ^a^		
						KF + 179 mM F		36.48 (4.63) ^a^		
						CaF2 + 6 mM F		33.06 (4.63) ^a^		
						CaF2 + 24 mM F		38.24 (4.63) ^a^		
						CaF2 + 179 mM F		0.88 (4.63) ^a^		
						Excite F		42.4 (4.63) ^a^		
	Yang et al., 2018 [39]	M (50)	Exp.	1% citric acid	Control	Control	10	46.5 ^b^ (4.63) ^a^	ER	µTBS
					CaP	CPP-ACP		41.2 ^b^ (4.63) ^a^		
						Novamin		31.4 ^b^ (4.63) ^a^		
SE + C	Zumstein et al., 2018 [51]	M (50)	Quasi-Exp.	pH cycling	Control	Control	20	24.7 (8.1) ^c^	SE	µTBS
					F	SnCl2/AmF4		16.3 (6.36) ^c^		
					Control	Control		15.43 (6.53) ^c^	Univ.	
					F	SnCl2/AmF4		14.12 (7.12) ^c^		
Storage in a fluid solution for 18-month measurement										
ER + B	Moreira et al., 2021 [15]	M (54)	Exp.	Cariogenic + S. Mutans	Control	Control	8	25.4 (2.45)	ER	µTBS
					F	NaF		7.81 (4.48)		
					CaP	CPP-ACP		26.01 (3.28)		
					Pept.	P11-4		25.24 (3.98)		

^a^—Input SD Values; ^b^—Information given by authors; ^c^—Information from another meta-analysis. Legend: B—Biological; C—Chemical; RoB—Risk of bias; ACAD—Artificial caries-affected dentin; BRP—Biomimetic remineralization procedure; SD—Standard deviation; AT—Adhesive technique; OM—Outcome measurement; ADS—Artificial demineralization solution; M—Medium; H—High; Exp.—Experimental; ER—Etch-and-Rinse; SE—Self-Etch; Univ.—Universal; F—Fluorine; Ca—Calcium; CaP—Calcium phosphate; Pept.—Peptide; FLs—Flavonoids; SiO_2_—Silica; Hap—Hidroxiapatite; HEMA—2-hydroxyethyl methacrylate; TMC—Thermocycling; µTBS—microtensile bond strength; SBS—shear bond strength; µSBS—microshear bond strength.

### 3.3. Meta-Regressions

#### 3.3.1. Influence of the Adhesive Technique on NMA Effect Estimates

The meta-regression results showed that the ER technique performed better than the SE in four NMA comparisons: control vs. calcium phosphate, control vs. peptide, fluorine vs. calcium phosphate, and fluorine vs. peptide. On the contrary, the SE technique performed better in the NMA comparison of peptide vs. hydroxyapatite. In all other comparisons, both techniques demonstrated similar performance (Appendix A Table A5).

#### 3.3.2. Influence of the ACAD Protocol on NMA Effect Estimates

Regarding the influence of different ACAD protocols on NMA effect estimates, the chemical ACAD protocol resulted in higher bond strength values than the biological ACAD protocol in nine NMA comparisons: control vs. fluorine, control vs. calcium phosphate, control vs. peptide, control vs. HEMA, control vs. flavonoids, control vs. calcium, control vs. hydroxyapatite, fluorine vs. calcium phosphate, and fluorine vs. peptide. In all other comparisons, both protocols performed similarly (Appendix A Table A6).

### 3.4. Network Meta-Analysis

Plots for the three performed NMAs are shown in Table 2.

Table 3 shows the NMA results from the BR intervention network.

The contribution tables are displayed in Appendix A, Table A7, Table A8 and Table A9.

#### 3.4.1. ER Technique with Chemical ACAD Protocol

The results of this NMA suggested that no statistically significant differences existed between any BR interventions in any of the network comparisons.

#### 3.4.2. ER Technique with Biological ACAD Protocol

When the ER technique and the biological ACAD protocol were used together, 8 of the 10 BR intervention network comparisons achieved statistically significant results: the calcium phosphate intervention compared to control (MD: −21.209, 95% CI: −25.954, −16.463), flavonoids (MD: −12.771, 95% CI: −20.538, −5.003), and fluorine (MD: −17.012, 95% CI: −22.103, −11.920); the flavonoids intervention compared to control (MD: 8.438, 95% CI: 2.289, 14.587); the peptide intervention compared to control (MD: 18.295, 95% CI: 14.418, 22.172), flavonoids (MD: 9.857, 95% CI: 2.588, 17.126), and fluorine (MD: 14.098, 95% CI: 9.684, 18.512); and the fluorine intervention compared to control (MD:4.197, 95% CI: 1.080, 7.314).

#### 3.4.3. SE Technique with Chemical ACAD Protocol

When the SE technique and the chemical ACAD protocol were used together, only 2 of the 36 BR intervention network’s comparisons achieved statistically significant results: the calcium phosphate (MD: −4.455, 95% CI: −8.857, −0.053) and the flavonoids (MD: −7.520, 95% CI: −14.758, −0.281) interventions compared to hydroxyapatite.

### 3.5. NMA Confidence Ratings

The confidence ratings for each NMA can be found in Appendix A, Table A10, Table A11 and Table A12.

#### 3.5.1. ER Technique with Chemical ACAD Protocol

In this NMA, two direct comparisons (calcium vs. control and control vs. fluorine) and one indirect comparison (hydroxyapatite vs. peptide) presented very low confidence, mainly due to major imprecision, heterogeneity, or incoherence concerns. The remaining indirect and direct comparisons presented a low or moderate confidence rating.

#### 3.5.2. ER Technique with Biological ACAD Protocol

In this NMA, all the direct and indirect comparisons presented a moderate confidence rating.

#### 3.5.3. SE Technique with Chemical ACAD Protocol

A low confidence rating was observed for six direct comparisons (calcium phosphate vs. peptide, calcium vs. fluorine, control vs. HEMA, control vs. SiO_2_, fluorine vs. HEMA, and fluorine vs. peptide) and two indirect ones (calcium vs. hydroxyapatite and HEMA vs. peptide), mostly due to major concerns in heterogeneity, incoherence, and within-study bias. The remaining comparisons presented a moderate confidence rating.

### 3.6. Rankings

The treatment rankings and probability of ranking best are displayed in Table 4.

#### 3.6.1. ER Technique with Chemical ACAD Protocol

Among all the treatments in the NMA, hydroxyapatite achieved the highest probability of being the best treatment (46.10%), closely followed by peptide (41.55%).

#### 3.6.2. ER Technique with Biological ACAD Protocol

In this NMA, calcium phosphate ranked first, with an 85.24% probability of being the best BR treatment.

#### 3.6.3. SE Technique with Chemical ACAD Protocol

Compared to the other treatments in the NMA, flavonoids achieved the highest probability of being best (46.36%), followed by HEMA (17.49%).

### 3.7. Sensitivity Analyses

The sensitivity analyses for each NMA can be found in Appendix A, Table A13, Table A14 and Table A15.

#### 3.7.1. ER Technique with Chemical ACAD Protocol

Both sensitivity analyses showed results like those of the main analysis.

#### 3.7.2. ER Technique with Biological ACAD Protocol

In this NMA, a sensitivity analysis where studies measuring the outcome with SBS tests were excluded was impossible because none used this test to assess the outcome. In the sensitivity analysis where we randomly selected one treatment intervention instead of combining interventions from the same category, the flavonoids vs. peptide comparison result lost statistical significance due to the loss of precision.

#### 3.7.3. SE Technique with Chemical ACAD Protocol

When we excluded studies using SBS tests from the NMA, the flavonoids vs. hydroxyapatite comparison ceased to show differences between the two interventions due to a loss of precision. When we randomly selected 1 treatment intervention instead of combining interventions from the same category, 8 of the 36 NMA comparison conclusions changed from not showing differences between the interventions to favoring one of them.

### 3.8. Discussion

This systematic review aimed to unravel the intricate interactions among different BR procedures and their influence on bond strength in human ACAD by analyzing and comparing bond strength from various in vitro studies through NMA. NMA allows for the integration of data from direct and indirect comparisons, enabling a more precise estimation of treatment effects and a deeper understanding of optimal treatment options. Ultimately, this systematic review and NMA aspires to contribute to the existing knowledge on dentin-bonding procedures and offer valuable insights into the effectiveness of BR. The findings may help clinicians make informed decisions regarding dentin-bonding strategies for improved treatment outcomes [51].

This study’s systematic review and NMA have shed light on the potential benefits of BR for bond strength in human ACAD, measured both immediately and after artificial aging. Its findings indicate that BR protocols are promising in enhancing restorative materials’ bonding performance on demineralized dentin surfaces. [52]

ACAD’s compromised nature negatively affects bond strength, and its surface is more challenging for bonding due to the incomplete infiltration of adhesives into the exposed collagen matrix [53]. Furthermore, the low pH associated with ACAD promotes the activation and activity of proteolytic enzymes, accelerating the breakdown of non-infiltrated collagen and the hybrid layer [37,48].

Our NMA findings highlighted differences between chemical and biological ACAD protocols. Chemical protocols consistently yielded higher bond strength results than biological, agreeing with previous research [54]. This difference may derive from the thicker demineralization layer associated with chemical protocols and the excessive softness of the primary dentine resulting from microbiological approaches [54].

The NMA also revealed variations in bond strength depending on the adhesive application technique. With their additional acid-etching stage, ER techniques proved more efficient in dissolving the smear layer than SE methods, which have a less acidic composition and are more sensitive [20]. Additionally, SE relies on chemical interactions with calcium ions, often found in lower concentrations in ACAD. Consequently, ER techniques yielded significantly higher bond strength values than SE, in line with the existing literature [31,33,53,55]. Moreover, when considering the ACAD surface, ER consistently demonstrated higher bond strength than SE materials [53].

This systematic review’s 23 in vitro studies showed medium heterogeneity, reflecting variations in ACAD protocols, aging methods, and bond strength tests. Thus, random-effects models were employed throughout the NMA investigation. Artificial aging methods, such as thermocycling and months of storage, generally reduce bond strength. Thermocycling promoted the most extreme breakdown of the bond interface and caused the lowest bond strength, even with associated BR, which is consistent with other studies [56]. However, different bond strength tests were used in the included investigations, which could affect the measurement results, and aspects such as specimen preparation and geometry, loading configuration, and material characteristics were not considered [3,57,58].

BR overall increased the bond strength values, even after artificial aging methods [10,58]. Nonetheless, the limited availability of studies reporting BR associated with bond strength restricts the exploration of these relationships [22]. Incorporating these BR methods into dental treatments can potentially enhance the durability and quality of the resin–dentin interface, offering promising avenues for improving clinical outcomes in restorative dentistry. In the NMA on ER with chemical ACAD, hydroxyapatite was the most effective treatment (46.10%), closely followed by peptide (41.55%), despite the low confidence in some comparisons. In the NMA on ER with biological ACAD, calcium phosphate emerged as the top-ranking BR (85.24%), significantly surpassing the control, flavonoids, and fluoride treatments. However, the NMA on SE with chemical ACAD showed low confidence in various comparisons, with flavonoids having the highest probability (46.36%) of being more effective, followed by HEMA (17.49%). These findings highlight the nuanced effectiveness of BR, influenced by different protocols and compositions. Most investigations on BR have shown its ability to remineralize ACAD in a basic manner. However, because they were carried out in vitro, their application in clinical contexts remains unexplored [22].

This study has some limitations. Most notably, in vitro studies lack the complexity of the oral environment, including oral biofluids and microbial interactions [3,22,52,56,57]. The absence of real dental caries development processes in the ACAD models is also a limitation. Future studies should address these shortcomings for a more comprehensive understanding of the clinical applicability of BR.

Another limitation is related to the sensitivity analysis for the NMA on SE with chemical ACAD. In this network, when we randomly selected one treatment intervention instead of combining interventions from the same category, 8 out of the 36 NMA comparisons changed their conclusions from not showing differences between the interventions to favoring one of them. Despite this, we are confident that combining multiple arms related to the same intervention yields more reliable estimates because it does not waste useful data and evidence, as outlined and in accordance with the Cochrane recommendations. Moreover, regardless of the strategy used to cope with multiple-arm trials, six of the eight comparisons that had their conclusions changed in the sensitivity analysis were based solely on indirect evidence, which inherently carries less confidence than scenarios where direct evidence is also available. 

Despite these limitations, our findings suggest that BR can enhance bond strength in ACAD, offering potential benefits for clinical practice. Dental professionals can use this knowledge to optimize treatment approaches, improve patient outcomes, and extend the longevity of adhesive bonding materials [3,22,52,57,58]. Future research should include randomized clinical trials to confirm the findings.

## 4. Conclusions

In conclusion, through a systematic review and NMAs, we showed that bond strength degraded after biological or chemical ACAD protocols. As a result, surface preparation with BR procedures prior to bonding is advised to increase the bonding of ER and SE adhesives.

## Figures and Tables

**Figure 1 ijms-26-03488-f001:**
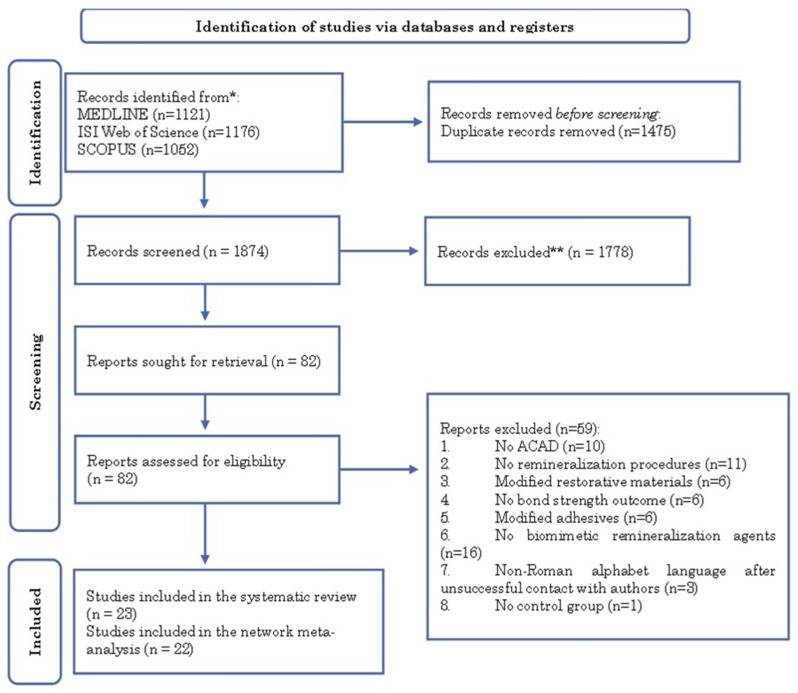
PRISMA 2020 flow diagram of literature search for new systematic reviews [18]. Identification *; screening **.

**Table 2 ijms-26-03488-t002:** Network meta-analysis plots.

**Plot of the NMA** **ER + Chemical ACAD**	**Plot of the NMA** **ER + Biological ACAD**	**Plot of the NMA** **SE + Chemical ACAD**
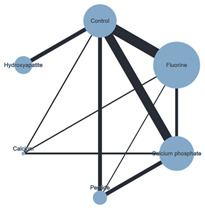	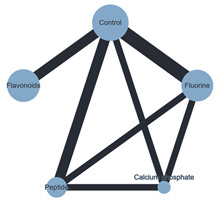	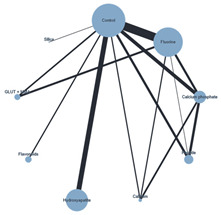

Note: Black lines connect biomimetic remineralization interventions that were compared head-to-head. The size of each node (circle) provides a measure of the sample size. The thickness of the line provides a measure of the number of direct comparisons between two interventions. Legend: ACAD—Artificial caries-affected dentin; ER—Etch-and-Rinse; NMA—Network meta-analysis; SE- Self-Etch.

**Table 3 ijms-26-03488-t003:** Network meta-analysis results from the network of biomimetic remineralization interventions.

NMA	NMA Results															
ER + chemical	Calcium															
	0.596 (−7.289, 8.482)		Calcium Phosphate													
	−0.508 (−8.207, 7.191)		−1.105 (−4.828, 2.619)			Control										
	−0.628 (−8.504, 7.248)		−1.224 (−5.931, 3.482)			−0.120 (−3.783, 3.544)			Fluorine							
	4.333 (−5.240,13.906)		3.736 (−3.063, 0.536)			4.841 (−0.848,10.530)			4.960 (−1.806,11.727)			HAp				
	4.044 (−4.923, 3.011)		3.448 (−1.833, 8.729)			4.553 (−0.635, 9.740)			4.672 (−1.320,10.665)			−0.288 (−7.988, 7.411)			Peptide	
ER + biological	Calcium Phosphate															
	−21.209 (−25.954, −16.463)			Control												
	−12.771 (−20.538, −5.003)			8.438 (2.289, 14.587)			Flavonoids									
	−17.012 (−22.103, −11.920)			4.197 (1.080, 7.314)			−4.241 (−11.135, 2.652)				Fluorine					
	−2.914 (−8.210, 2.382)			18.295 (14.418, 22.172)			9.857 (2.588, 17.126)				14.098 (9.684, 18.512)			Peptide		
SE + chemical	Calcium															
	2.663 (−2.395, 7.722)	Calcium Phosphate														
	1.124 (−3.670, 5.917)	−1.539 (−4.817, 1.738)			Control											
	5.728 (−2.442,13.897)	3.065 (−4.318, 10.447)			4.604 (−2.011, 11.219)	Flavonoids										
	0.523 (−4.358, 5.404)	−2.140 (−5.787, 1.506)			−0.601 (−2.932, 1.730)	−5.205 (−12.219, 1.809)		Fluorine								
	3.023 (−3.815, 9.861)	0.360 (−5.589, 6.309)			1.899 (−3.210, 7.009)	−2.705 (−11.063, 5.654)		2.500 (−2.603, 7.603)		HEMA						
	−1.792 (−7.415, 3.831)	−4.455 (−8.857, −0.053)			−2.916 (−5.854, 0.023)	−7.520 (−14.758, −0.281)		−2.315 (−6.066, 1.436)		−4.815 (−10.709, 1.079)		HAp				
	2.654 (−3.216, 8.524)	−0.009 (−4.076, 4.058)			1.530 (−2.373, 5.434)	−3.074 (−10.755, 4.607)		2.131 (−2.213, 6.475)		−0.369 (−6.728, 5.990)		4.446 (−0.440, 9.332)	Peptide			
	1.964 (−5.802, 9.730)	−0.699 (−7.633, 6.234)			0.840 (−5.270, 6.950)	−3.764 (−12.769, 5.241)		1.441 (−5.099, 7.980)		−1.059 (−9.024, 6.906)		3.756 (−3.024, 10.536)	−0.690 (−7.941, 6.560)			Silica

Note: The data in each cell are the mean difference with 95% confidence intervals for the network comparison of row-defining treatment versus column-defining treatment. Negative values favor the intervention in the column. Statistically significant results are in bold and gray. Legend: ER—Etch-and-Rinse; SE—Self-Etch; HAp—Hydroxyapatite; HEMA—2-hydroxyethyl methacrylate.

**Table 4 ijms-26-03488-t004:** Treatment rankings and probability of ranking best.

NMA	Ranks and Probability of Ranking Best				
ER + chemical		Rank			
		Mean	Median	CrI95%	Probability of ranking best (%)
	Control	4.66	5	(3.6)	0.05^−4^
	Fluorine	4.66	5	(2.6)	0.64
	CaP	3.75	4	(2.6)	1.75
	Peptide	1.92	2	(1.5)	41.55
	Calcium	4.06	4	(1.6)	9.91
	HAp	1.97	2	(1.5)	**46.10**
ER + biological		Rank			
		Mean	Median	CrI95%	Probability of ranking best (%)
	Control	4.98	5	(5.5)	0.00
	Fluorine	3.89	4	(4.5)	0.00
	CaP	1.15	1	(1.2)	**85.24**
	Peptide	1.86	2	(1.2)	14.56
	FLs	3.12	3	(3.4)	0.20
SE + chemical		Rank			
		Mean	Median	CrI95%	Probability of ranking best (%)
	Control	5.49	6	(3.6)	0.11
	Fluorine	5.96	6	(3.9)	0.32
	CaP	3.28	3	(1.7)	12.41
	Peptide	4.77	4	(1.9)	5.01
	Calcium	6.00	7	(1.9)	4.15
	HAp	7.89	8	(4.9)	0.09
	FLs	2.75	2	(1.9)	**46.36**
	HEMA	4.03	3	(1.9)	17.49
	Silica	4.85	5	(1.9)	14.05

Note: Interventions ranked best are highlighted in bold. Legend: ER—Etch-and-Rinse; SE—Self-Etch; CrI—Credible interval; CaP—Calcium phosphate; FLs—Flavonoids, HAp—Hydroxyapatite; HEMA—2-hydroxyethyl methacrylate.

## Data Availability

The data that support the findings of this study are available from the corresponding author, Rosário Costa, upon reasonable request.

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
