# Peer review of "Biomimetic Remineralization Strategies for Dentin Bond Stability—Systematic Review and Network Meta-Analysis"

_ijms, 2025, doi:10.3390/ijms26083488_

Round 1
Reviewer 1 Report
Comments and Suggestions for Authors
I recommend the review publication.
Reason for overall recommendation: It is an interesting study and will be a useful tool for researchers to quickly find a comparison between the data in the literature, the methods required for a study in the field and the references. From my point of view, this review will save them a lot of the work of searching and detecting references.
This review makes a statistical interpretation of the data from the existing literature in the field of Biomimetic Dentin Remineralization of Tooth Enamel.
This review is relevant to the field because it can be a starting point for future research. The data analyzed by the authors are well structured in tables and diagrams and will help young researchers.
The assessment of the accuracy of the techniques and the confidence rating are subjective
Storing a large number of data in a few graphs. (compared with other published material)
Specific improvements should the authors consider regarding the methodology
I took an IBM-style statistics course and have some knowledge in the field but I think a statistician can give you more information. But from the point of view of collecting and selecting articles it seems ok.
the conclusions consistent with the evidence and arguments presented
the references appropriate
Author Response
Dear Reviewer,
Thank you for your comments, which we find pertinent and very useful.
1 - "The assessment of the accuracy of the techniques and the confidence rating are subjective."
Answer: We acknowledge that the assessment of technique accuracy and confidence ratings involves some subjectivity. However, we followed consistent criteria and relied on prior literature to minimize bias.
2 - "Storing a large number of data in a few graphs."
Answer: To ensure clarity and focus on the main points, we chose to present the core data in the main graphs and include the additional data in the appendix. This approach allowed us to highlight the key findings while still providing full access to all relevant information for comparison.
3 - "Specific improvements should the authors consider regarding the methodology"
Answer: We tried to shorten a bit the methodology in the text and chose not to go into further detail, due to number of words constrains.
4 - "but I think a statistician can give you more information"
Answer: the methodology has been validated by an experienced statistician in this kind of research, ensuring the adequacy of our methodological approach.
Reviewer 2 Report
Comments and Suggestions for Authors
The presented work concerns an interesting issue from the point of view of materials science and its potential clinical applications. The introduction is well prepared, but I lack a hypothesis that can be formulated in the case of a systematic review.
The methodology has been described in detail, and I have no comments on it.
The results are described in detail and presented clearly in the form of tables. No reservations. The discussion is detailed and concerns the interpretation of the results against the background of available knowledge, but also broadly discusses the limitations associated with the results of the analyses presented in the assessed work. This part of the article is adequate for the results obtained and does not require any changes.
The conclusions are the result of the conducted analyses. They are adequate for the results.
Therefore, interesting and very well prepared work, they recommend accepting the work for publication.
Author Response
Dear Reviewer,
Thank you for your positive feedback.
1-“ The introduction is well prepared, but I lack a hypothesis that can be formulated in the case of a systematic review.”
Answer: In the case of a systematic review, our approach was to synthesize existing evidence rather than test a specific hypothesis. However, we acknowledge the importance of a clearly defined research question and have structured our study to systematically address it.
Reviewer 3 Report
Comments and Suggestions for Authors
--Main Address of this Review---
The meta-analysis aimed to evaluate the bond strength of artificial caries-affected dentin of permanent human teeth with and without biomimetic remineralization , assessed in in vitro studies.
--Reason for Overall Recommendation---
According to the analysis of the results, the conclusion is considered valid as it is supported by consistent data, a robust methodology, and alignment with the existing scientific literature.
well-founded interpretations, connects the study's findings with existing literature, highlighting important contributions, adequately reflects the results obtained and addresses the proposed objectives.
--Comment for Make it more impactful/Improvement---
1. After this analysis of all the presented studies, it would be interesting to "conduct randomized clinical trials to confirm the findings", assess the effectiveness of the interventions, and ensure the applicability of the results in clinical practice
2. The references are apt; however, there are no references between 2024 and 2025
--Overall Detail---
Topic of discussion is new and relevant to the area of study as it involves one of the biggest issues of dentistry: the recurrence of caries.
The study explores the methods of remineralization to prevent recurrence by examining each of the related aspects in detail.
The topic of discussion is of great importance as it encapsulates and provides available literature, which makes it easy to analyze the result achieved.
Furthermore, the study fills a specific gap since it investigates the bond strength between two popular adhesive systems.
Finally, the study is very clinically applicable, despite the limitation of being based on in vitro investigations.
----Table/Data comments--
The tables are easy to understand, with descriptive headings and captions that clearly define their meaning. They serve well to facilitate comprehension of the findings.
The data are correctly organized and standardized, with proper units, and the tables are properly referenced in the text.
--Language ---
Language is clear and precise, comprehension, well-prepared graphs and tables and contribute to understanding the results.
Author Response
Dear Reviewer,
Thank you for your insightful comments, which we find pertinent and very useful.
1 - "After this analysis of all the presented studies, it would be interesting to "conduct randomized clinical trials to confirm the findings", assess the effectiveness of the interventions, and ensure the applicability of the results in clinical practice."
Answer: We appreciate your suggestion regarding the need for randomized clinical trials. We agree that conducting such studies would be valuable in confirming our findings, assessing intervention effectiveness, and ensuring their applicability in clinical practice. We can emphasize this point more clearly in our discussion. “Future research should include randomized clinical trials to confirm the findings.”- added at the end of discussion.
2- "The references are apt; however, there are no references between 2024 and 2025"
Answer: Regarding the references, we acknowledge the lack of citations from 2024 and 2025. Given the rapidly evolving nature of research, we did a review on the latest literature to incorporate any relevant and recent studies that strengthen our work.